

# A hybrid model to simulate the annual runoff of Kaidu River in northwest China

**J. Xu[1], Y. Chen[2], L. Bai[1], and Y. Xu[1]**

[1]The Research Center for East–West Cooperation in China, School of Geographic Sciences, East China Normal University, Shanghai 200241, China
[2]State Key Laboratory of Desert and Oasis Ecology, Xinjiang Institute of Ecology and Geography, Chinese Academy of Sciences, Urumqi 830011, China

Received: 9 December 2015 – Accepted: 11 December 2015 – Published: 18 January 2016

Correspondence to: J. H. Xu (jhxu@geo.ecnu.edu.cn)

Published by Copernicus Publications on behalf of the European Geosciences Union.

**HESSD**

doi:10.5194/hess-2015-529

A hybrid model to simulate the AR of Kaidu River, China

J. Xu et al.



**Abstract**

The fluctuant and complicated hydrological process result in the uncertainty of runoff forecast. Thus, the multi-method integrated modeling approaches to simulate runoff are necessary. Integrating the ensemble empirical mode decomposition (EEMD), the back propagation artificial neural network (BPANN) and nonlinear regression equation, we put forward a hybrid model to simulate the annual runoff (AR) of the Kaidu River in northwest China. We also validated the simulated effect by using the coefficient of determination (CD) and Akaike information criterion (AIC) based on the observed data from 1960 to 2012 in Dashankou hydrological station. The average absolute and relative error show the high simulation accuracy of the the hybrid model. CD and AIC both illustrate that the simulated effect of the hybrid model is much better than that of a single BPANN. The hybrid model and integrated approach elicited by this study may be applied to simulate the annual runoff of the similar rivers in northwest China.

## 1   Introduction

Hydrologic models are the basis of hydrological simulation. Commonly used hydrologic models contain three types, i.e. stochastic models, dynamics models and distributed models.

Because the hydrology process is closely interconnected with climatic process, and greatly influenced by climate change, the key of hydrological simulation is to understand the climatic- hydrological process. Therefore, stochastic models and dynamics models all focus on climatic- hydrological process.

The stochastic hydrologic models are black box systems, based on data and using mathematical and statistical concepts to link a certain input (such as rainfall, evaporation, temperature, etc.) to the output (such as runoff, etc.). Commonly used techniques are regression, transfer functions, neural networks and system identification.

**HESSD**

doi:10.5194/hess-2015-529

**A hybrid model to simulate the AR of Kaidu River, China**

J. Xu et al.

The dynamics hydrologic models try to represent the physical processes observed in the real world. Typically, such models contain representations of surface runoff, subsurface flow, evapotranspiration, and channel flow, but they can be far more complicated. These models are known as deterministic hydrology models. Deterministic hydrology models can be subdivided into single-event models and continuous simulation models. Distributed hydrologic models are grid-cell based and take into account the spatial variability of meteorological input and other inputs like terrain, soils, vegetation and land use. In distributed hydrological models, runoff generated in a grid cell is transported downstream through a grid cell to grid cell network using the local drain direction of each grid cell.

Among the three types of hydrologic model, the distributed hydrological model is most popular and widely used because the mechanism of producing process of runoff was well described by the model. As we all know, SWAT (Soil and Water Assessment Tool) and SWAP (Soil Water–Atmosphere–Plant) are the two popular distributed hydrological models all over the world. SWAT model is a continuation of nearly 30 years of modelling efforts conducted by the USDA Agricultural Research Service, and widely used in the word. A number of scientists used SWAT model for simulating streamflow and related hydrologic analyses (Gan and Luo, 2013; Levesque et al., 2008; Liu et al., 2008, 2014; Luo et al., 2012; Shope et al., 2014; Lin et al., 2015). According to the investigation by Gassman et al. (2007), hundreds of articles had been identified that report SWAT applications, reviews of SWAT components, or other research that includes SWAT. physically based land surface model SWAP has been intensively validated during the last 10 years (van Dam et al., 1997; Gusev and Nasonova, 2003; Kroes et al., 2003; Gusev et al., 2011; Ma et al., 2011). Different versions of SWAP were validated against various observed hydrothermal characteristics. The validations were performed for "point" experimental sites and for catchments and river basins of different areas (from $10^{-1}$ to $10^{5}$ km$^2$) on a long-term basis and under different natural conditions (Nasonova and Gusev, 2007).

**HESSD**

doi:10.5194/hess-2015-529

**A hybrid model to simulate the AR of Kaidu River, China**

J. Xu et al.

Discussion Paper | Discussion Paper | Discussion Paper | Discussion Paper |

However, the application prerequisite of the distributed hydrological model is to successfully obtain a large number of parameters (such as temperature, precipitation, evapotranspiration, topography, land use, soil moisture, vegetation coverage, etc.) at each grid cell. But for a large river basin with sparse meteorological and hydrological sites as well as lacking observed data, it is difficult to obtain the large number of parameters at each grid cell. Therefore, more studies are required to explore the hydrological process from different perspectives and using different methods.

For understanding the variation pattern of streamflow, hydrologists used various simulation methods and models. In the recent 20 years, various methods such as grey model (Yu et al., 2001; Trivedi and Singh, 2005), functional-coefficient time series model (Shao et al., 2009), wavelet analysis (Labat et al., 2000a, b; Lane, 2007; Sang, 2012), genetic algorithm (Seibert, 2000), artificial neural network (Hsu et al., 1995; Hu et al., 2008; Tokar and Johnson, 1999; Modarres, 2009) have been widely applied for runoff simulation. Specially, hybrid models (Nourani et al., 2009; Zhao et al., 2009; Sahay and Srivastava, 2014; Xu et al., 2014; Yarar, 2014) have been paid more attention.

The water resource in northwest China which can be utilized is mainly from the streamflow of inland rivers. So the runoff variation of an inland river has aroused more and more attention (Chen et al., 2009; Li et al., 2008; Wang et al., 2010; Xu et al., 2011). However, the runoff variation pattern of inland rivers in northwest China has not been clearly comprehended because of the complexity of the hydrological process (Xu et al., 2009, 2010). To understand the runoff variation pattern of inland rivers in northwest China, this study selected the Kaidu River as a typical representative of an inland river in northwest China, and integrated the ensemble empirical mode decomposition (EEMD), the back propagation artificial neural network (BPANN) and nonlinear regression equation to conduct a hybrid model for simulating annual runoff (AR).

**HESSD**

doi:10.5194/hess-2015-529

**A hybrid model to simulate the AR of Kaidu River, China**

J. Xu et al.

## 2 Study basins and data

### 2.1 Study area

The Kaidu River is situated at the north fringe of Yanqi Basin on the south slope of the Tianshan Mountains in Xinjiang, and is enclosed between latitudes 42°14′–43°21′ N and longitudes 82°58′–86°05′ E (Fig. 1). The river starts from the Hargat Valley and the Jacsta Valley in Sarming Mountain with a maximum altitude of 5000 m (the middle part of the Tianshan Mountains), and ends in Bosten Lake, which is located in the Bohu County of Xinjiang. This lake is the biggest lake in Xinjiang (also once the biggest interior fresh water lake in China) and immediately starts another river known as the Kongque River. The catchment area of the Kaidu River above Dashankou, is 18 827 km$^2$, with an average elevation of 3100 m (Chen et al., 2013).

Bayanbuluke wetland, which is in Kaidu River Basin, is the largest wetland of the Tianshan Mountain area. The large areas of grassland and marshes in Bayanbuluke wetland have provided favorable conditions for swan survival and reproduction. For this reason, it becomes the China's sole state-level swan nature reserve. The annual average temperature is only −4.6 °C and the extreme minimum temperature is −48.1 °C. The snow cover days are as many as 139.3 d and the largest average snow depth is 12 cm. As the unique high alpine cold climate and topography, it cultivates various alpine grassland and meadow ecosystems, having abundant aquatic plants, animals and good grassland resources. It is the birthplace and water saving place of the Kaidu River and plays a crucial role in regulating, reserving water and maintaining water balance. It also plays the utmost important role in protecting the Bosten Lake, its surrounding wetlands, and the ecological environment and green corridor of the lower reaches of Tarim River.

**HESSD**

doi:10.5194/hess-2015-529

**A hybrid model to simulate the AR of Kaidu River, China**

J. Xu et al.

## 2.2 Data

Because the human activity Kaidu River Basin is rare, the Dashankou hydrological station is located at the outfall in front mountain areas where the amount of water used by humans is negligible compared to the total discharge. Therefore, it can be assumed that the observed data of runoff reflect the natural conditions (Chen et al., 2013).

The purpose of this study is to well understand the internal variation pattern by simulation method, so we used the annual runoff (AR) time series data from 1960 to 2012, which are observed at the Dashankou hydrological station.

## 3 Methodology

To simulate the AR, we made a hybrid model by integrating ensemble empirical mode decomposition (EEMD), back propagation artificial neural network (BPANN) and regression equation. We firstly used the EEMD method to decompose the AR to four intrinsic mode functions (i.e. IMF1, IMF2, IMF3 and IMF4) and a trend (RES). Then we simulated IMFs by the back propagation artificial neural network (BPANN), and simulated RES (trend) by a nonlinear regression equation. Finally, the simulated values for AR are obtained from the summation of the simulated results of the trend (RES) and IMFs. The framework of the hybrid model is showed as in Fig. 2.

## 3.1 EEMD method

The ensemble empirical mode decomposition (EEMD) is a new noise-assisted data analysis method based on the empirical mode decomposition (EMD), which defines the true IMF components as the mean of an ensemble of trials, each consisting of a signal plus white noise of finite amplitude (Wu and Huang, 2009).

The EMD has been developed for non-linear and non-stationary signal analysis, though only empirically. The EMD decompose a signal into several intrinsic mode functions (IMFs), then the frequencies of the IMFs are arranged in decreasing order

Discussion Paper | Discussion Paper | Discussion Paper | Discussion Paper | Discussion Paper |

**HESSD**

doi:10.5194/hess-2015-529

**A hybrid model to simulate the AR of Kaidu River, China**

J. Xu et al.

(high to low), where the lowest frequency of the IMF components represents the overall trend of the original signal or the average of the time series data (Huang et al., 1998). Most importantly, each of these IMFs must satisfy two conditions: (1) the number of extrema and the number of zero crossings must be equal or differ at most by one; (2) at any point, the mean value of the envelope defined by the local maxima and local minima must be zero.

The EMD processing is as follows.

For the original signal $x(t)$, first find out all the local maxima and minima, and then use cubic spline interpolation method to form the upper envelope $u_1(t)$ and the lower envelope $u_2(t)$; the local mean envelope $m_1(t)$ can be expressed as:

$$m_1(t) = \frac{1}{2}(u_1(t) + u_2(t)) \tag{1}$$

The first component $h_1(t)$ can be obtained by subtracting the local mean envelope $m_1(t)$ from the original signal $x(t)$, with the mathematical expression as follows:

$$h_1(t) = x(t) - m_1(t) \tag{2}$$

If $h_1(t)$ does not satisfy the IMF conditions, regard it as the new $x(t)$, and repeat the steps in formula (Eqs. 1 and 2) $k$ times until $h_{1k}(t)$ is obtained as an IMF.

$$h_{1k}(t) = h_{1(k-1)}(t) - m_{1k}(t) \tag{3}$$

Designate $C_1 = h_{1k}$, and select a stoppage criterion defined as follows:

$$SD = \sum_{t=0}^{T} \left[ \frac{h_{1(k-1)}(t) - h_{1k}(t)}{h_{1(k-1)}(t)} \right]^2 \tag{4}$$

Here, the standard deviation (SD) is smaller than a predetermined value. If the above process is repeated too many times, the IMF will become a pure frequency modulation

Discussion Paper | Discussion Paper | Discussion Paper | Discussion Paper | Discussion Paper |

**HESSD**

doi:10.5194/hess-2015-529

**A hybrid model to simulate the AR of Kaidu River, China**

J. Xu et al.

signal with constant amplitude in the actual operation, possibly resulting in loss of its actual meaning.

Once the first IMF component is determined, the residue $r_1(t)$ can also be obtained by separating $C_1$ from the rest of the data, i.e.

$$r_1(t) = x(t) - C_1 \tag{5}$$

By taking the residue $r_1(t)$ as new data and repeating steps (Eqs. 1–5), a series of IMFs, namely, $C_2, C_3, \cdots, C_n$ can be obtained.

The sifting process finally stops when the residue, $r_n(t)$, becomes a monotonic function or a function with only one extremum from which no more IMF can be extracted. Finally, the original signal $x(t)$ can be reconstructed by $n$ IMFs (i.e. $C_i(t)$) and a residue $r_n(t)$ as follows:

$$x(t) = \sum_{i=1}^{n} C_i(t) + r_n(t) \tag{6}$$

Although EMD has many merits, there is a shortcoming of mode mixing in EMD. To overcome the mode mixing problem, the EEMD has been developed for non-linear and non-stationary signal analysis (Wu and Huang, 2009).

The principle of EEMD is that: adding white noise to the data, which distributes uniformly in the whole time–frequency space, the bits of signals of different scales can be automatically designed onto proper scales of reference established by the white noise.

The EEMD algorithm is straightforward and can be described as follows: first, add a white noise series to the original signal

$$x_i(t) = x(t) + n_i(t) \tag{7}$$

Where $x_i(t)$ is the new signal after adding $i$th white noise to the original signal data $x(t)$, $n_i(t)$ is the white noise. Then, decompose the signal with added white noise into

Discussion Paper | Discussion Paper | Discussion Paper | Discussion Paper

**HESSD**

doi:10.5194/hess-2015-529

**A hybrid model to simulate the AR of Kaidu River, China**

J. Xu et al.

IMFs using EMD according to the steps of Eqs. (1)–(5) equation, the corresponding IMF components $C_{ij}(t)$ and residue component $r_i(t)$ of the decompositions were obtained. Finally, adopt the means of the ensemble corresponding to the IMFs of the decompositions as the final result, namely

$$C_j(t) = \frac{1}{N} \sum_{i=1}^{N} C_{ij}(t) \tag{8}$$

where $C_j(t)$ is the final $j$th IMF component, $N$ is the number of white noise series, $C_{ij}(t)$ denotes the $j$th IMF from the added white noise trial.

Wu and Huang (2009) noted that the amplitude size of the added noise exerts little influence on the decomposition results on the condition that it is limited, is not vanishingly small or very large and can include all possibilities. Therefore, the application of the EEMD method does not rely on subjective involvement; it is an adaptive data analysis method.

The significance test in EEMD can be carried out by means of white noise ensemble disturbance, to get each IMF credibility (Wu and Huang, 2009; Huang and Shen, 2005).

In addition, to solve the overshooting and undershooting phenomenon of the impact of the boundary on the decomposition process, mirror-symmetric extension (Huang and Shen, 2005; Xue et al., 2013) was used to address the EEMD decomposition boundary problem.

The residue of EEMD is a monotonic function that intrinsically presents the overall trend of a time series (Wu et al., 2007, 2009). Thus, the reconstruction of signal $x(t)$ based on EEMD can be obtained as following:

$$x(t) = \text{IMF1} + \text{IMF2} + \cdots + \text{IMF}_n + \text{RES (trend)} \tag{9}$$

Where RES is the residue of EEMD, i.e. the trend of signal $x(t)$.

In this study, we decomposed the AR time series to a trend (RES) and four IMFs.

**HESSD**

doi:10.5194/hess-2015-529

**A hybrid model to simulate the AR of Kaidu River, China**

J. Xu et al.

## 3.2 BPANN

In the back propagation artificial neural network (BPANN), a number of smaller processing elements (PEs) are arranged in layers: an input layer, one or more hidden layers, and an output layer (Hsu et al., 1995). The input from each PE in the previous layer ($x_i$) is multiplied by a connection weight ($w_{ji}$). These connection weights are adjustable and may be likened to the coefficients in statistical models. At each PE, the weighted input signals are summed and a threshold value ($\theta_j$) is added. This combined input ($I_j$) is then passed through a transfer function ($f(\cdot)$) to produce the output of the PE ($y_j$). The output of one PE provides the input to the PEs in the next layer. This process can be summarized (Maier and Dandy, 1998) in Eqs. (13) and (14) and illustrated in Fig. 3.

$$I_j = \sum w_{ji} x_i + \theta_j \tag{10}$$

$$y_i = f(I_j) \tag{11}$$

The error function of network at $t$th moment is defined as follows:

$$E(t) = \frac{1}{2} \sum_{j=1}^{q} [y_j(t) - d_j(t)]^2 \tag{12}$$

where $y_i(t)$ is the actual output and $d_i(t)$ is the desired output respectively corresponding to $i$th neuron at $t$th moment. When $E(t) \leq \varepsilon$ ($\varepsilon$ is a given error in advance), the network will stop training and the network model at this time is just what we need.

We used the BPANN with a four-tier structure to simulate IMF1, IMF2, IMF3 and IMF4 of the AR based on the results from the EEMD. The four-tier structure of the BPANN for each IMF is as follows (Fig. 4): an input layer with three variables, i.e. $(t-1)$th, $(t-2)$th and $(t-3)$th value of the IMF; two hidden layers, in which the first layer contains three neurons and the second layer contains four neurons; an output layer with a variable, i.e. $t$th value of the IMF.

Discussion Paper | Discussion Paper | Discussion Paper | Discussion Paper |

**HESSD**

doi:10.5194/hess-2015-529

**A hybrid model to simulate the AR of Kaidu River, China**

J. Xu et al.

The transfer function from the input layer to two hidden layers is tansig, i.e. the hyperbolic tangent sigmoid transfer function (http://www.mathworks.com/help/nnet/ref/tansig.html). The transfer function from the hidden layers to the output layer is purelin, i.e. the linear function (http://www.mathworks.com/help/nnet/ref/purelin.html).

The purpose of our BPANN is to capture the relationship between a historical set of inputs and corresponding outputs. As mentioned above, this is achieved by repeatedly presenting examples of the input/output relationship to the model and adjusting the model coefficients (i.e. the connection weights) in an attempt to minimize an error function between the historical outputs and the outputs predicted by the model. This calibration process is generally referred to as "training". The aim of the training procedure is to adjust the connection weights until the global minimum in the error surface has been reached. The network training process (Moghadassi et al., 2009) is summarized in Fig. 5.

The back-propagation process is commenced by presenting the first example of the desired relationship to the network. The input signal flows through the network, producing an output signal, which is a function of the values of the connection weights, the transfer function and the network geometry. The output signal produced is then compared with the desired (historical) output signal with the aid of an error (cost) function.

Because it can train any network as long as its weight, net input, and transfer functions have derivative functions (Kermani et al., 2005), we selected the Levenberg–Marquardt optimization, i.e. trainlm (http://www.mathworks.com/help/nnet/ref/trainlm.html) as a network training function in the computing environment of MATLAB.

## 3.3 Nonlinear regression

In order to simulate the trend of AR, we fitted a quadratic polynomial by using the nonlinear regression based on the results from the EEMD. We conducted the quadratic

**HESSD**

doi:10.5194/hess-2015-529

**A hybrid model to simulate the AR of Kaidu River, China**

J. Xu et al.

polynomial regression equation as follows:

$$y = at^2 + bt + c \tag{13}$$

where the independent variable ($t$) is the time variable, and the dependent variable ($y$) represent the trend of AR, which is the RES obtained from the EEMD. The coefficients ($a$, $b$ and $c$) are obtained by method of least squares (Lancaster and Šalkauskas, 1986).

## 3.4 Simulated effect test

In order to identify the uncertainty of the simulated results, the coefficient of determination was calculated as follows:

$$\mathrm{CD} = 1 - \frac{\mathrm{RSS}}{\mathrm{TSS}} = 1 - \frac{\sum\limits_{i=1}^{n}(y_i - \hat{y}_i)^2}{\sum\limits_{i=1}^{n}(y_i - \overline{y})^2} \tag{14}$$

where CD is the coefficient of determination; $\hat{y}_i$ and $y_i$ are the simulate value and actual data of AR respectively; $\overline{y}$ is the mean of $y_i (i = 1, 2, \ldots, n)$; $\mathrm{RSS} = \sum\limits_{i=1}^{n}(y_i - \hat{y}_i)^2$ is the residual sum of squares; $\mathrm{TSS} = \sum\limits_{i=1}^{n}(y_i - \overline{y})^2$ is the total sum of squares. The coefficient of determination is a measure of how well the simulate results represent the actual data. A bigger coefficient of determination indicates a higher certainty and lower uncertainty of the estimates (Xu, 2002).

To compare the goodness between our hybrid model and single BPANN, we also used the measure of Akaike information criterion (AIC) (Anderson et al., 2000). The formula of AIC is as follows:

$$\mathrm{AIC} = 2k + n\ln(\mathrm{RSS}/n) \tag{15}$$

**HESSD**

doi:10.5194/hess-2015-529

**A hybrid model to simulate the AR of Kaidu River, China**

J. Xu et al.

Interactive Discussion

Discussion Paper | Discussion Paper | Discussion Paper | Discussion Paper

where $k$ is the number of parameters estimated in the model; $n$ is the number of samples; RSS is the same as in formula (Eq. 14). A smaller AIC indicates a better model (Burnham and Anderson, 2002).

## 4  Results and discussion

### 4.1  Decomposition for AR

Figure 6 reveals anomaly fluctuations of the AR time series in the Kaidu River during 1960–2012. It is clear that the AR shows a strong nonlinear and non-stationary variation. Because of the nonlinear and non-stationary characteristics, it is difficult to show the change law of the AR time series.

To discover intrinsic modes in the signal of AR, we decomposed the AR time series by the EEMD method. For decomposing the AR time series, the ensemble number is 100, and the added noise has amplitude that is 0.2 times the standard deviation of the corresponding data, and four IMF components (IMF1–4) and a trend component (RES) were obtained. The decomposed results are showed in Fig. 7. Each IMF component reflects an intrinsic mode with its fluctuation characteristics. The IMFs reflects the fluctuation characteristics from high frequency to low frequency. IMF1 presents the most high frequency fluctuation, IMF4 with lowest frequency fluctuation. Whereas the fluctuation frequency of IMF2 is higher than that of IMF3 but lower than that of IMF1, and the fluctuation frequency of IMF3 is higher than that of IMF4 but lower than that of IMF2. The residue (RES) of EEMD is a monotonic function that presents the overall trend of the AR time series.

The significance test showed that IMF2, IMF3 and IMF4 reach above the 95 % confidence level, while IMF1 reach above 90 % confidence level. The variance contribution rate of IMF1, IMF2, IMF3, IMF4 and RES (trend) is 28.29, 19.61, 10.11, 8.58 and 33.41 % respectively. The summation of IMF1, IMF2, IMF3, IMF4 and RES represent of the reconstruction for AR time series, which is very highly correlative with

Discussion Paper | Discussion Paper | Discussion Paper | Discussion Paper |

**HESSD**

doi:10.5194/hess-2015-529

**A hybrid model to simulate the AR of Kaidu River, China**

J. Xu et al.

**HESSD**

doi:10.5194/hess-2015-529

**A hybrid model to simulate the AR of Kaidu River, China**

J. Xu et al.

its original data series. It can be seen that the reconstruction for AR series with the original data series is almost exactly the same (see Fig. 8). This result illustrates that the decomposition of the AR time series by EEMD got a good prospective effect.

## 4.2 Simulation for IMFs

In order to capture the relationship between the historical data and real time output, we constructed the BPANN with a four-tier structure to simulate IMF1, IMF2, IMF3 and IMF4 of the AR based on the results from the EEMD. Using the MATLAB software (http://www.mathworks.com/products/matlab/), we respectively selected the transfer function for input layer to the hidden layer and the hidden layer to the output layer as the tangent sigmoid function (tansig) and the linear function (purelin), and chose "trainlm" as a training function to train the network. We set the learning rate as 0.01 and the training error accuracy as 0.01, and randomly extracted 70, 15 and 15 % of the data in the time series of each IMF as the training, testing, and validation samples. We finally obtained the optimized network for each IMF after thousands of training. Using the optimized networks, we obtained the simulated results for IMF1, IMF2, IMF3 and IMF4 respectively (Fig. 9). The coefficient of determination (CD) indicates that the simulated accuracy for each IMF is very high (Table 1). Of course, the simulated effect of each IMF is different. The AIC (Table 1) values indicate that the simulated effect of IMF4 is the best, and then follows are IMF3, IMF2, and IMF1 respectively.

## 4.3 Simulation for the trend

As above mentioned, the residue (RES) of EEMD presents the overall trend of the AR time series. Because it is a monotonic function, we can simulate the trend by a regression equation. Based on the data of RES from EEMD, we obtained the regression equation by using the method of least squares as the following quadratic

polynomial:

$$y = 0.002t^2 - 7.7975t + 7632.6 \tag{16}$$

Where $t$ is the time, which is measured by year; and $y$ is the simulated value for the trend of the AR time series.

The coefficient of determination of formula (Eq. 16) is as high as 0.9999. It is evident that the simulated effect of the RES (trend) is even better than that of IMF1, IMF2, IMF3 and IMF4 (also see Table 1). The simulated results for the trend of AR time series calculated by formula (Eq. 16) are shown as Fig. 10.

## 4.4 Simulation for AR

Based on the idea and framework of the hybrid model mentioned in the methodology of this study, we can calculate the simulated value of AR at each year by summing the simulated value of IMF1, IMF2, IMF3, IMF4 and RES.

By summing the simulated value of IMF1, IMF2, IMF3, IMF4 and RES at each year, we calculated the simulated value of AR for each year. The simulated values with original data of the AR are shown as Fig. 11.

In order to compare and validate the simulated results from the hybrid model, we also simulated the AR series by using a single BPANN. Table 2 shows the simulated effect comparisons between the hybrid model and the single BPANN. It can be seen that the coefficient of determination (CD) of the hybrid model is high as 0.9772, whereas that of the single BPANN is only 0.4037. Moreover, the AIC value of the hybrid model (0.7390) is far smaller than that of the single BPANN (171.7801). It is clear that both CD and AIC value indicate that the simulated effect of the hybrid model is much better than that of the single BPANN. Furthermore, the average absolute and relative error show the high simulation accuracy of the the hybrid model. All the indices illustrate that the hybrid model much better that a single BPANN.

**HESSD**

doi:10.5194/hess-2015-529

**A hybrid model to simulate the AR of Kaidu River, China**

J. Xu et al.

## 5 Conclusions

Integrating the ensemble empirical mode decomposition, the back propagation artificial neural network and nonlinear regression equation, we conducted a hybrid model to simulate the annual runoff of the Kaidu River in northwest China. The main conclusions of this study are as follows:

1. The comparison between simulated values of annual runoff and its original data shows the high simulation accuracy of the hybrid model. Both of the small average absolute and relative errors illustrate the high simulation accuracy of the hybrid model. The big CD and small AIC both indicate that the simulated effect of the hybrid model is much better than that of a single back propagation artificial neural network.

2. This study elicited an integrated approach to simulate annul runoff of inland rivers, and the framework of the hybrid model conducted by this study may be applied to the other inland rivers in northwest China.

*Acknowledgements.* This work was supported by State Key Laboratory of Desert and Oasis Ecology, Xinjiang Institute of Ecology and Geography, Chinese Academy of Sciences (Program No: Y371163).

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

**Table 1.** CD and AIC of simulatiing models for the IMFS and trend of AR.

| IMFS | CD | AIC |
|------|--------|-----------|
| IMF1 | 0.9107 | 0.5789 |
| IMF2 | 0.9619 | −54.9342 |
| IMF3 | 0.9859 | −105.9041 |
| IMF4 | 0.9980 | −204.2977 |
| Trend | 0.9999 | −405.1425 |

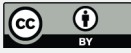

**HESSD**

doi:10.5194/hess-2015-529

**A hybrid model to simulate the AR of Kaidu River, China**

J. Xu et al.

Discussion Paper | Discussion Paper | Discussion Paper | Discussion Paper | Discussion Paper



**Table 2.** Simulated effect comparisons between the hybrid model and single BPANN.

| IMFS | Hybrid model | Single BPANN |
|---|---|---|
| CD | 0.9772 | 0.4037 |
| AIC | 0.7390 | 171.7801 |
| Average absolute error ($10^8 \, \mathrm{m}^3$) | 0.8099 | 3.5477 |
| Average relative error (%) | 2.3313 | 10.1079 |

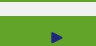

**HESSD**

doi:10.5194/hess-2015-529

**A hybrid model to simulate the AR of Kaidu River, China**

J. Xu et al.

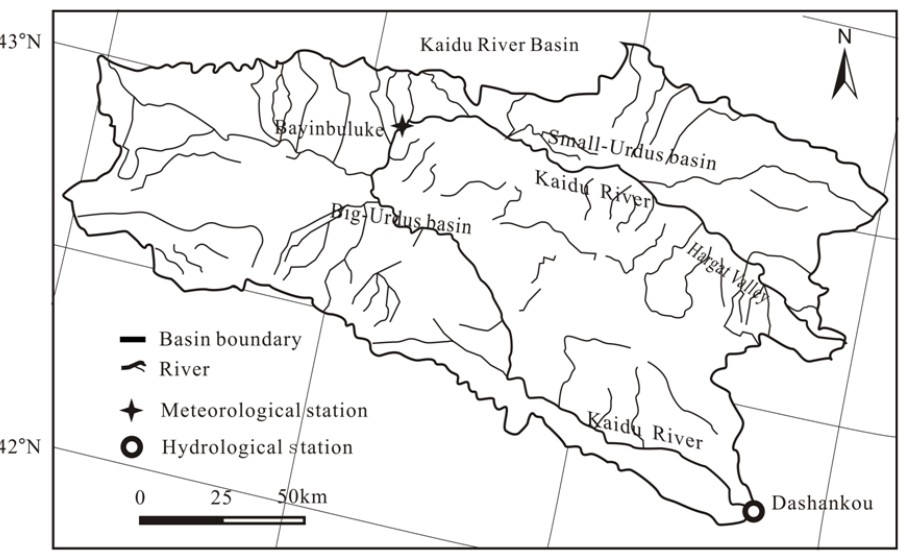

**Figure 1.** Location of the Kaidu River, northwest China.

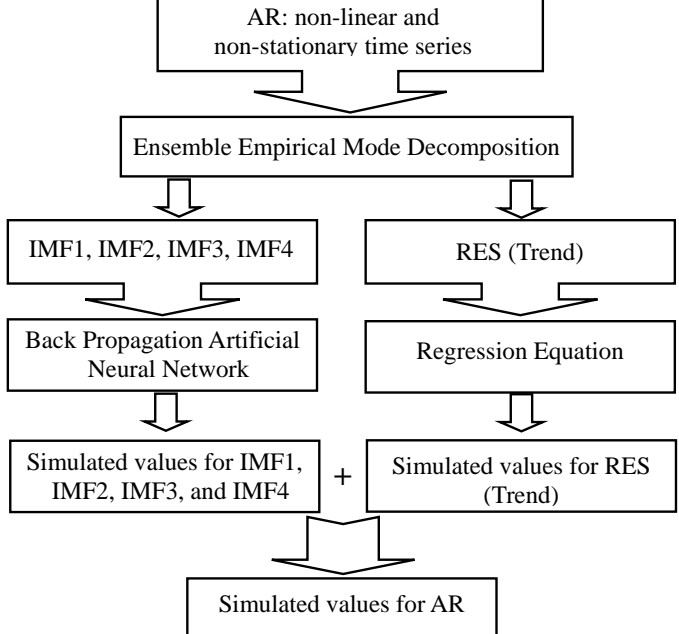

**Figure 2.** The framework of the hybrid model to simulate the annual runoff.

HESSD

doi:10.5194/hess-2015-529

**A hybrid model to simulate the AR of Kaidu River, China**

J. Xu et al.

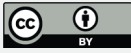

**HESSD**

doi:10.5194/hess-2015-529

**A hybrid model to simulate the AR of Kaidu River, China**

J. Xu et al.

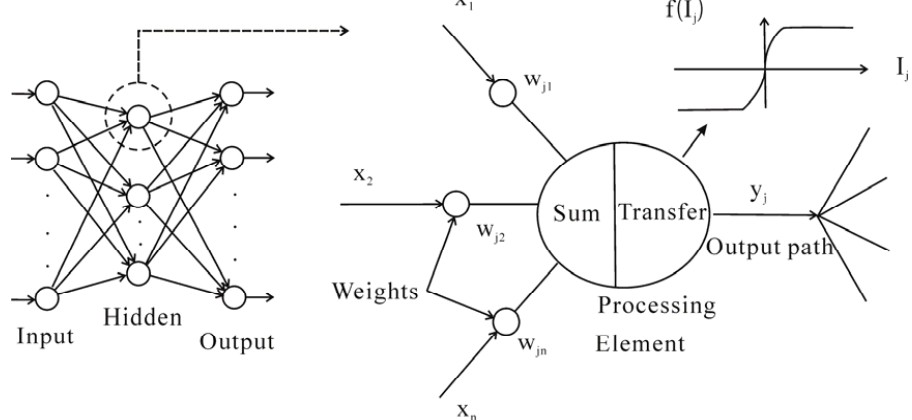

**Figure 3.** The back-propagation artificial neural network.

**HESSD**

doi:10.5194/hess-2015-529

**A hybrid model to simulate the AR of Kaidu River, China**

J. Xu et al.

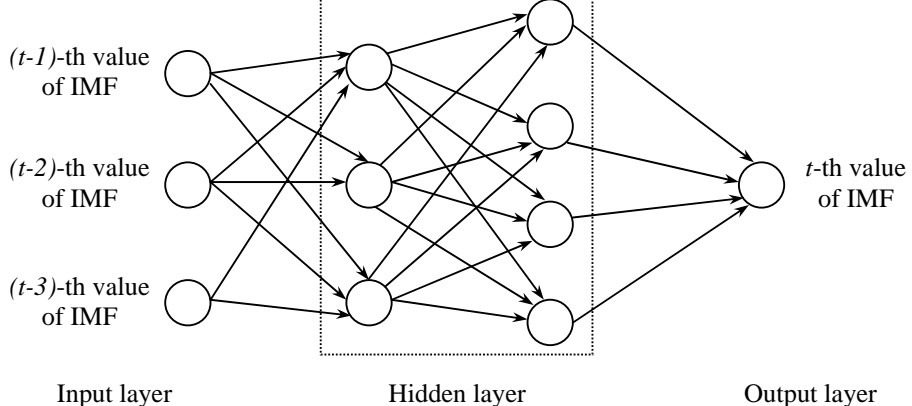

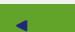 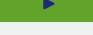
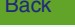 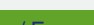

**Figure 4.** Four-tier structure BPANN to simulate the IMFs of AR.

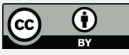

Discussion Paper | Discussion Paper | Discussion Paper | Discussion Paper

**HESSD**

doi:10.5194/hess-2015-529

**A hybrid model to simulate the AR of Kaidu River, China**

J. Xu et al.



```
┌─────────────────┐
│ Initialize training │
│    epoch =1       │
└─────────────────┘
         │
         ▼
┌─────────────────┐
│ Initial weights and biases │
│  with random values │
└─────────────────┘
         │
         ▼
┌─────────────────┐
│ Present input pattern and │
│  calculate output values │
└─────────────────┘
         │
         ▼
┌─────────────────┐
│   Calculate MSE   │
└─────────────────┘
         │
         ▼
      ◇ MSE ≤ ε ◇ ──── Yes ────┐
         │ No                    │
         ▼                       ▼
epoch = epoch + 1 ◄── ◇ epoch ≥ epoch_max ◇ ── Yes ──► Stop training network
         ▲              │ No
         │              ▼
         └──── Update weights and biases
```

**Figure 5.** Back-propagation training process.

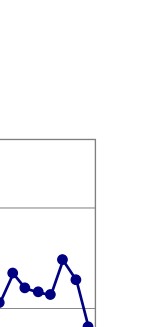

**Figure 6.** Annual runoff anomalies in the Kaidu River during 1960–2012.

Discussion Paper | Discussion Paper | Discussion Paper | Discussion Paper | Discussion Paper |

**HESSD**

doi:10.5194/hess-2015-529

**A hybrid model to simulate the AR of Kaidu River, China**

J. Xu et al.



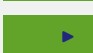

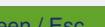

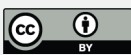

**Figure 7.** The EEMD results for the time series of AR in the Kaidu River.

HESSD

doi:10.5194/hess-2015-529

A hybrid model to simulate the AR of Kaidu River, China

J. Xu et al.

Title Page

Abstract · Introduction

Conclusions · References

Tables · Figures

◀ · ▶︎

◀ · ▶

Back · Close

Discussion Paper | Discussion Paper | Discussion Paper | Discussion Paper

Discussion Paper | Discussion Paper | Discussion Paper | Discussion Paper

**HESSD**

doi:10.5194/hess-2015-529

**A hybrid model to simulate the AR of Kaidu River, China**

J. Xu et al.

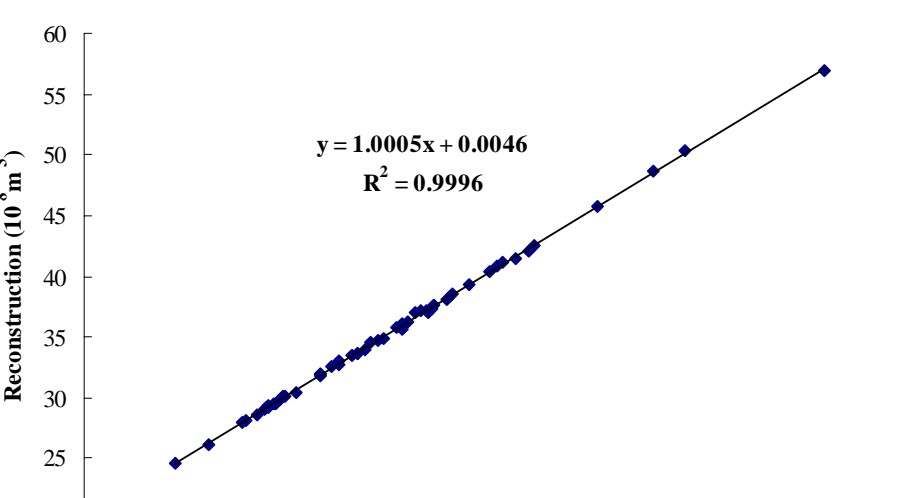

**Figure 8.** The correlation between the reconstruction of AR time series based on EEMD and its original data.

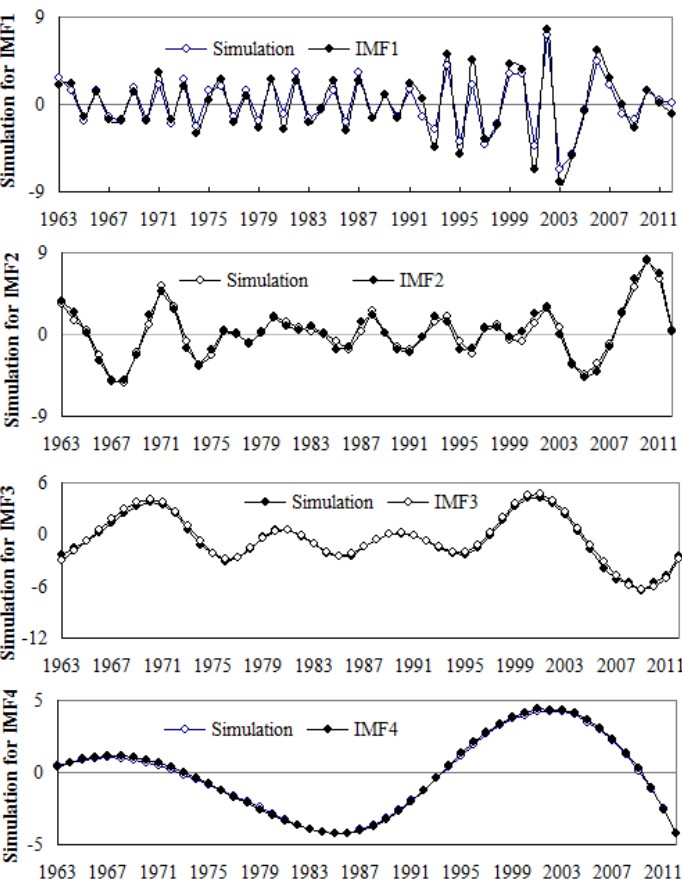

**Figure 9.** Simulation for the IMFS by BPANN.

**HESSD**

doi:10.5194/hess-2015-529

**A hybrid model to simulate the AR of Kaidu River, China**

J. Xu et al.

Discussion Paper | Discussion Paper | Discussion Paper | Discussion Paper

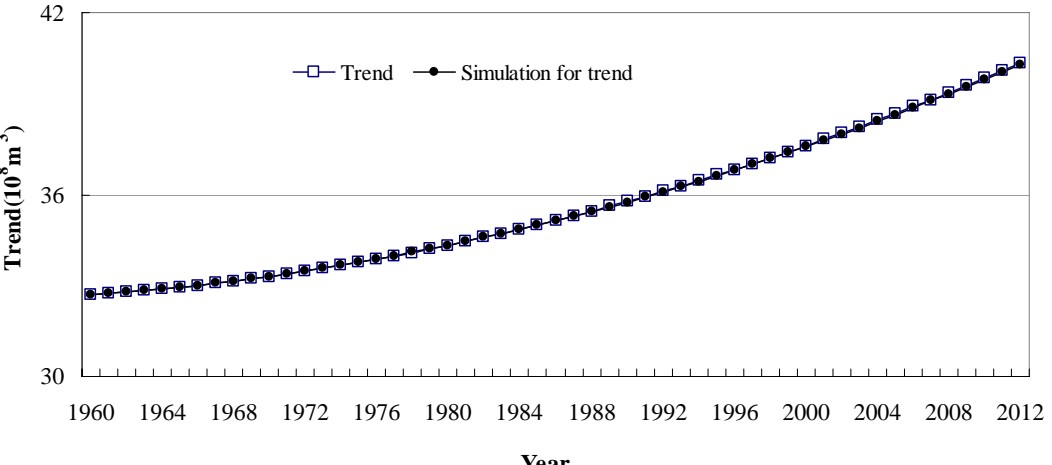

**Figure 10.** Simulation for the trend by regreesion equation.

Discussion Paper | Discussion Paper | Discussion Paper | Discussion Paper |

**HESSD**

doi:10.5194/hess-2015-529

**A hybrid model to simulate the AR of Kaidu River, China**

J. Xu et al.



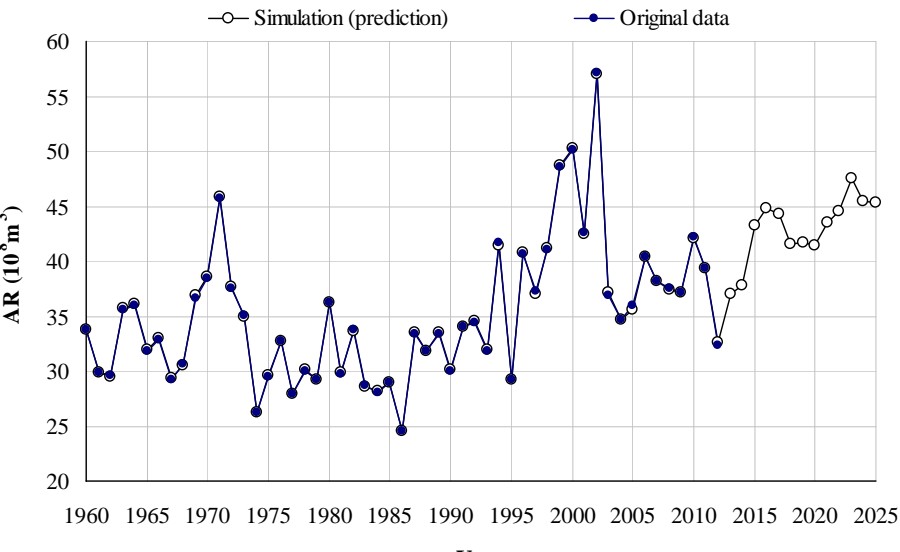

**Figure 11.** Simulation (prediction) of AR by the hybrid model.

HESSD

doi:10.5194/hess-2015-529

A hybrid model to simulate the AR of Kaidu River, China

J. Xu et al.

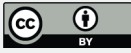