# Peer review of "A hybrid model to simulate the annual runoff of Kaidu River in northwest China"

_Hydrology and Earth System Sciences, 2015_

## Referee Comment (RC1) · Anonymous Referee #1 · 1 Feb 2016

The ensemble empirical mode decomposition (EEMD) is integrated with a back propagation artificial neural network (BPANN) and a nonlinear regression equation for simulating annual runoff for a watershed. My major comments are: 1) Figure 11 compares simulated and observed annual runoff. My suggestion is to break the time period with observation (1960-2012) into calibration and validation periods (e.g., 1960-1989 for calibration and 1990-2012 for validation). The calibration period is used for parameter estimation for the EEMD, BPANN and nonlinear regression equation. 2) Figure 7: the physical meanings for MF1, MF2, MF3, MF4 and Trend components need to be explained. MF1∼MF4 are corresponding to different frequencies. Which kinds of climatic phenomena are corresponding to each of the components (e.g., El Niño and La Niña)? What's the meaning of the trend and its contributing factors (e.g., land use change etc)? 3) For the BPANN, what are the inputs each of the components (MF1∼MF4)? Does

these inputs vary from MF1 to MF4? This may be related to the second comments.

I think the manuscript needs some general revision of the English language. For examples, a few minor comments are listed below:

Line 15 on page 2: "...contain three types, i.e. stochastic models, dynamics models and distributed models." Please revise this since stochastic VS deterministic, lumped vs distributed, conceptual VS physically-based. Correspondingly, the first paragraph on page 3 may need to be revised.

Lines 20-21 on page 2: "Therefore, stochastic models and dynamics models all focus on climatic- hydrological process." The logic is unclear. Please revise this paragraph.

Line 24 on page 2: "evaporation" is usually an output. "potential evaporation" is an input.

Line 22 on page 3: "physically based land surface..."

———————————————————

---

## Referee Comment (RC2) · Anonymous Referee #2 · 5 Feb 2016

General comments:

Combining the advantages of EEMD and BPANN, this study conducted a hybrid model to simulate the annual runoff (AR) of the Kaidu River, northwest China. The approach and idea of this study may be referenced to the similar study. I think the topic will be of interest for readers and the manuscript deserves publication. However, the manuscript needs a minor revision before its publication (see below).

Specific comments:

1. Line 9 of page 6, the sub-title "3 Methodology" should be changed to "3 Methods". 2. In order to facilitate the readers to iterate the computing process, the MATLAB program for EEMD should be indicated by denoting the related references in the section "3.1 EEMD method". 3. Generally, the coefficient of determination is denoted as R2 in

statistics. The CD in formula (14) on page 12 seems to be the $R^2$. Please confirm that the CD is the same meaning with $R^2$. If my understanding correct, they should be unified. The related question is that the $R^2$ value of the formula (16) on page 15 should be marked out. 4. The section "4 Results and discussion" needs more discussions. The authors should give an explanation why the hybrid model is much better than a single BPANN. What is the reason for this? 5. To avoid any error, please carefully check all words and sentences in the whole text before the manuscript to be resubmitted again. For example: (1) Line 22 of page 3, the first alphabet of the first word in the sentences "physically based land surface model. . . . . ." should be capital, i.e. "physically" should be change to "Physically". (2) Line 24~25 of page 15, the sentences "All the indices illustrate that the hybrid model much better that a single BPANN" should be changed to "All the indices illustrate that the hybrid model is much better than a single BPANN".

---

## Author Comment (AC1) · 26 Feb 2016

Dear Anonymous Referee #1,

We would like to thank you for your constructive and useful comments on how to improve our manuscript. We have already revised our paper according to your comments. Attached you will find the answers to your observations.

With kind regards,

Yours sincerely,

Authors, Jianhua Xu, Yaning Chen, Ling Bai, Yiwen Xu

2016-02-06

[Figure]

Please also note the supplement to this comment:
http://www.hydrol-earth-syst-sci-discuss.net/hess-2015-529/hess-2015-529-AC1-supplement.pdf

—————————————————————

[Figure]

**Supplement:**

Dear Anonymous Referee #1,

We would like to thank you for your constructive and useful comments on how to improve our manuscript. We have already revised our paper according to your comments. The details are as follows.

**Comment 1:** Figure 11 compares simulated and observed annual runoff. My suggestion is to break the time period with observation (1960-2012) into calibration and validation periods (e.g., 1960-1989 for calibration and 1990-2012 for validation). The calibration period is used for parameter estimation for the EEMD, BPANN and nonlinear regression equation.

**Response of authors:** Yes, we revised that according to your suggestion. For calibration and validation purposes, we divided the whole data series into two periods, the calibration period, i.e. 1960-1989, and the validation period, i.e. 1990-2012. The calibration period is used for parameter estimation for the EEMD, BPANN and nonlinear regression equation. The validation period is used for validating the effectiveness of the hybrid model. The simulation results show the excellent performances of the model for both the calibration (1960-1989) and validation (1990-2012) periods with $R2$ and AIC value (Table 3), which is highly acceptable. Fig. 12 shows the observed data of AR and its simulated values by the hybrid model.

It should be noted that we inserted a new figure in our revised paper (i.e. Fig. 9), and the original figure 11 in the primary manuscript was change to figure 12.

[Figure]

Figure 12    Comparisons between the observed data of AR and its simulated values for calibration period (1960–1989) and validation period (1990–2012)

**Comment 2:** Figure 7: the physical meanings for MF1, MF2, MF3, MF4 and Trend components need to be explained. MF1~ MF4 are corresponding to different frequencies. Which kinds of climatic phenomena are corresponding to each of the components (e.g., El Niño and La Niña)? What's the meaning of the trend and its contributing factors (e.g., land use change etc)?

**Response of authors:** Yes, each IMF component in Fig. 7 has its own physical meaning, which reflects the inherent oscillation at a characteristic scale. The four IMF components (IMF1-4) reflect the fluctuation characteristics from high frequency to low frequency. IMF1 presents the most high frequency fluctuation, IMF4 with lowest frequency fluctuation. Whereas the fluctuation

[revised manuscript text omitted]

**Comment 3:** For the BPANN, what are the inputs each of the components (MF1~MF4)? Does these inputs vary from MF1 to MF4?
**Response of authors:** The four-tier structure of the BPANN for each IMF is as follows (Fig. 4): an input layer with three variables, i.e. $(t-1)$-th, $(t-2)$-th and $(t-3)$-th value of the IMF; two hidden layers, in which the first layer contains three neurons and the second layer contains four neurons; an output layer with a variable, i.e. $t$th value of the IMF.

**Comment 4:** Line 15 on page 2: ": : :contain three types, i.e. stochastic models, dynamics models and distributed models." Please revise this since stochastic VS deterministic, lumped vs distributed, conceptual VS physically-based. Correspondingly, the first paragraph on page 3 may need to be revised. Lines 20-21 on page 2: "Therefore, stochastic models and dynamics models all focus on climatic- hydrological process." The logic is unclear. Please revise this paragraph.
**Response of authors:** Yes, we revised these according to your suggestion. Please see page 2~3. The descriptions in the revised paper are as following: the description of hydrological processes is the basis of hydrological modelling and simulation. Many different types of models have been developed for describing hydrological processes during the past decades. These hydrologic models

can be classified as stochastic and deterministic models according to their mathematical property, or classified as conceptual and physically based models according to the physical processes involved in modelling, or classified as lump and distributed models according to the spatial description of the watershed process (Refsgaard, 1996; Moglen and Beighley, 2002).

**Comment 5:** I think the manuscript needs some general revision of the English language.
**Response of authors:** Yes, According to the comments and suggestions, some grammars and spelling errors have been corrected, and the English has also been polished by one of my colleagues from America.

Again, we would like to thank you for your generous comments given to the improvement of our manuscript.

Best wishes,

Yours sincerely,

Authors,
Jianhua Xu, Yaning Chen, Ling Bai, Yiwen Xu

2016-02-06

---

## Author Response (AR1)

Dear editor, Professor Dawen Yang,

Thank you very much for your decision on our manuscript, which is helpful to us for improving the article's quality.

We also would like to thank the two anonymous referees for their constructive and useful comments on how to improve our manuscript. We have already revised our paper point by point according to their comments.

We have already submitted our responses to the review comments in the open review process, and now we reproduce them below for reference.

**1. Responses to the anonymous referee #1**

The details are as follows.

**Comment 1:** Figure 11 compares simulated and observed annual runoff. My suggestion is to break the time period with observation (1960-2012) into calibration and validation periods (e.g., 1960-1989 for calibration and 1990-2012 for validation). The calibration period is used for parameter estimation for the EEMD, BPANN and nonlinear regression equation.
**Response of authors:** Yes, we revised that according to your suggestion. For calibration and validation purposes, we divided the whole data series into two periods, the calibration period, i.e. 1960-1989, and the validation period, i.e. 1990-2012. The calibration period is used for parameter estimation for the EEMD, BPANN and nonlinear regression equation. The validation period is used for validating the effectiveness of the hybrid model. The simulation results show the excellent performances of the model for both the calibration (1960-1989) and validation (1990-2012) periods with R2 and AIC value (Table 3), which is highly acceptable. Fig. 12 shows the observed data of AR and its simulated values by the hybrid model.
It should be noted that we inserted a new figure in our revised paper (i.e. Fig. 9), and the original figure 11 in the primary manuscript was change to figure 12.

[Figure]

Figure 12  Comparisons between the observed data of AR and its simulated values for calibration period (1960–1989) and validation period (1990–2012)

**Comment 2:** Figure 7: the physical meanings for MF1, MF2, MF3, MF4 and Trend components need to be explained. MF1~ MF4 are corresponding to different frequencies. Which kinds of climatic phenomena are corresponding to each of the components (e.g., El Niño and La Niña)? What's the meaning of the trend and its contributing factors (e.g., land use change etc)?

**Response of authors:** Yes, each IMF component in Fig. 7 has its own physical meaning, which reflects the inherent oscillation at a characteristic scale. The four IMF components (IMF1-4) reflect the fluctuation characteristics from high frequency to low frequency. IMF1 presents the highest frequency fluctuation and IMF4 shows lowest frequency fluctuation. Whereas the fluctuation frequency of IMF2 is higher than that of IMF3 but lower than that of IMF1, and the fluctuation frequency of IMF3 is higher than that of IMF4 but lower than that of IMF2. The residue (RES) of EEMD is a monotonic function that presents the overall trend of the AR time series.

The multi-scale oscillations of runoff in the Kaidu River reflect not only the periodic changes of the climatic system under external forcing but also the non-linear feedback of the climatic system. To compare the hydrological cycle of Kaidu River and the El Niño meteorological phenomena, we also decomposed the NINO3.4 index data series in the same period by using the EEMD method. The result is that the four IMF components (IMF1-4) of the NINO3.4 index data series respectively display quasi-3-year, quasi-6-year, quasi-11-year and quasi-28-year periodic fluctuation (Fig. 9), whereas the four IMF components (IMF1-4) of the AR series in the Kaidu River respectively show quasi-3-year, quasi-6-year, quasi-11-year and quasi-27-year cyclic variation (Fig. 7). Although the two cycles are not complete same, they show some comparability. A study showed that there is a possible variability in droughts and wet spells over China on the multi-year or decadal scale when one strong El Niño event happens, but it does not mean that each El Niño event must cause a wet-dry change (Su and Wang, 2006). Similarly, the larger fluctuations of runoff in the Kaidu River on the multi-year or decadal scale possibly relate to strong El Niño events, but it does not mean that a big change of runoff certainly corresponds to a strong El Niño event. The possible reason is that the influencing factors include not only El Niño event but also other factors.

[Figure]

Figure 9    The EEMD results for the NINO3.4 index data series during 1960 – 2012
(a new inserted figure in our revised paper)

In fact, there are many other factors affecting the runoff, such as the varied topography, vegetation cover and construction of water conservancy project (Chen et al., 2013). Our previous study showed that the runoff process of the Kaidu River is closely related to the regional climate change (Xu, et al., 2014; Bai, et al., 2015). To compare the cycles between the runoff in Kaidu River and the regional climatic factors in the study period, we used the EEMD method to decompose the data series of annual precipitation (AP) and annual average temperature (AAT) to four IMF components (IMF1-4) and a trend. The results are similar to that of the AR: the AP and AAT on the whole show an upward trend, meanwhile, a) the AP presents quasi-3-year, quasi-6-year, quasi-11-year and quasi-27-year cycles, and b) the AAT displays quasi-3-year, quasi-6-year, quasi-13-year and quasi-27-year cycles. To further analyze the correlation between runoff and precipitation and temperature, we reconstructed inter-annual and inter-decadal precipitation and temperature variations, in which the inter-annual precipitation/temperature is obtained by IMF1 and IMF2, while the inter-decadal precipitation/temperature is obtained by IMF3 and IMF4. The

results of multi-scale correlation analysis among annual runoff, annual precipitation and annual average temperature are shown in Table 1 (a new inserted figure in our revised paper). Evidently, although there are differences in the length and strength of the periods among the precipitation, temperature and runoff changes, the positive correlation between runoff and precipitation, temperature are still significant except for inter-annual precipitation v.s. inter-decadal runoff, suggesting that the precipitation and temperature are both the main causes of runoff variation. Furthermore, the higher correlation between runoff and climate factors is precipitation, followed by temperature at both the inter-annual and inter-decadal scales.

Table 1 Correlations between runoff and climate factors
(A new inserted figure in our revised paper)

| Time scale | Precipitation vs. runoff | Temperature vs. runoff |
|---|---|---|
| Inter-annual scale | 0.666** | 0.416** |
| Inter-annual v.s. inter-decadal scale | 0.205 | 0.441** |
| Inter-decadal v.s. inter-annual scale | 0.279* | 0.438** |
| Inter-decadal scale | 0.822** | 0.617** |

Note: **correlation is significant at the 0.01 level (2-tailed); *correlation is significant at the 0.05 level (2-tailed).

**Comment 3:** For the BPANN, what are the inputs each of the components (MF1~MF4)? Does these inputs vary from MF1 to MF4?
**Response of authors:** The four-tier structure of the BPANN for each IMF is as follows (Fig. 4): an input layer with three variables, i.e. $(t-1)$-th, $(t-2)$-th and $(t-3)$-th value of the IMF; two hidden layers, in which the first layer contains three neurons and the second layer contains four neurons; an output layer with a variable, i.e. $t$th value of the IMF.

**Comment 4:** Line 15 on page 2: "......contain three types, i.e. stochastic models, dynamics models and distributed models." Please revise this since stochastic VS deterministic, lumped vs distributed, conceptual VS physically-based. Correspondingly, the first paragraph on page 3 may need to be revised. Lines 20-21 on page 2: "Therefore, stochastic models and dynamics models all focus on climatic- hydrological process." The logic is unclear. Please revise this paragraph.
**Response of authors:** Yes, we revised these according to your suggestion. Please see page 2~3. The descriptions in the revised paper are as following: The description of hydrological processes is the basis of hydrological modelling and simulation. Many models have been developed for describing hydrological processes over the past decades. From different perspectives, these hydrologic models can be classified as stochastic and deterministic models according to their mathematical property, or classified as conceptual and physically based models according to the physical processes involved in modelling, or classified as lump and distributed models according to the spatial description of the watershed process (Refsgaard, 1996; Moglen and Beighley, 2002).

**Comment 5:** I think the manuscript needs some general revision of the English language.
**Response of authors:** Yes, According to the comments and suggestions, some grammars and spelling errors have been corrected, and the English has also been polished by one of my colleagues from America.

**2. Responses to the anonymous referee** #2

The details are as follows.

**Comment 1:** Line 9 of page 6, the sub-title "3 Methodology" should be changed to "3 Methods".
**Response of authors:** Yes, we revised that according to your suggestion.

**Comment 2:** In order to facilitate the readers to iterate the computing process, the MATLAB program for EEMD should be indicated by denoting the related references in the section "3.1 EEMD method".
**Response of authors:** Yes, according to your suggestion, we gave the related references and the web site where the MATLAB program for EEMD can be downloaded. We indicated that the MATLAB programs for EEMD are provided by RCADA, National Central University, which can be downloaded at the website (http://rcada.ncu.edu.tw/research1_clip_ex.htm).

**Comment 3:** Generally, the coefficient of determination is denoted as $R^2$ in statistics. The CD in formula (14) on page 12 seems to be the $R^2$. Please confirm that the CD is the same meaning with $R^2$. If my understanding correct, they should be unified. The related question is that the $R^2$ value of the formula (16) on page 15 should be marked out.
**Response of authors:** Yes, you are right. We already confirmed that the coefficient of determination is denoted as $R^2$ in statistics, and the CD in our paper is the same meaning with $R^2$. In order to unify express, we already changed the "CD" to "$R^2$" in the whole text.

**Comment 4:** The section "4 Results and discussion" needs more discussions. The authors should give an explanation why the hybrid model is much better than a single BPANN. What is the reason for this?
**Response of authors:** Yes, we have done as your suggestion. We explained that the reason for "the hybrid model is better than a single BPANN" as follows: All the indices illustrate that the hybrid model is much better than a single BPANN. The reason for this is that the hybrid model concentrated the advantages of both EEMD and BPANN. Where the EEMD can precisely decompose the non-linear and non-stationary signal of AR to intrinsic mode functions (IMFs), and the BPANN can well recognize and accurately simulate the IMFs. Because the non-linear and non-stationary AR signal contains many components and each component has its own intrinsic mode, a single BPANN can not accurately recognized and simulated the all change patterns in AR series. For this reason, this study used an integrated approach to conduct the hybrid model. In order to identify the pattern of each component in the non-linear and non-stationary AR signal, we firstly used the EEMD to decompose the AR series to four intrinsic mode functions (i.e. IMF1, IMF2, IMF3 and IMF4) and a trend (RES). Then we used the BPANN to accurately recongnize the pattern of each IMF by net learning and training, while using the nonliner regression to exactly simulate the pattern of the trend (RES). The above simulated results have already proved that our hybrid model is effective.

**Comment 5:** To avoid any error, please carefully check all words and sentences in the whole text before the manuscript to be resubmitted again. For example: (1) Line 22 of page 3, the first alphabet of the first word in the sentences "physically based land surface model….." should be

capital, i.e. "physically" should be change to "Physically". (2) Line 24~25 of page 15, the sentences "All the indices illustrate that the hybrid model much better that a single BPANN" should be changed to "All the indices illustrate that the hybrid model is much better than a single BPANN".

**Response of authors:** Yes, According to the comments and suggestions, some grammars and spelling errors have been corrected, and the English has also been polished by one of my colleagues from America.

Additionally, we also added some references as follows.

Bai, L., Chen, Z. S, Xu, J. H., and Li, W. H.: Multi-scale response of runoff to climate fluctuation in the headwater region of Kaidu River in Xinjiang of China, Theor. Appl. Climatol., DOI: 10.1007/s00704-015-1539-2, 2015.

Moglen, G. E. and Beighley, R. E.:   Spatially explicit hydrologic modeling of land use change, Journal of the American Water Resources Association, 38, 241-253, 2002.

Refsgaard, J.C.: Terminology, modelling protocol and classification of hydrologic model codes, in: Abbott, M. B. and Refsgaard, J. C. (Eds.), Distributed Hydrologic Modelling, Kluwer Academic Publishers, dordrecht, 41-54, 1996.

Su, M. F. and Wang, H. J.: Relationship and its instability of ENSO Chinese variations in droughts and wet spells, Sci China Ser D-Earth Sci., 50, 145-152, 2007.

Yang, D. W. and Musiake, K.: A continental scale hydrological model using the distributed approach and its application to Asia, Hydrol. Process., 17, 2855-2869, 2003.

Yang, D. W., Gao, B., Jiao, Y., Lei, H. M., Zhang, Y. L., Yang, H. B., and Cong, Z. T.: A distributed scheme developed for eco-hydrological modeling in the upper Heihe River, Sci China-Earth Sci., 58, 36-45, 2015.

Again, we would like to thank you and the anonymous referees for the hard work on our manuscript.

Best wishes,

Yours sincerely,

Authors,
Jianhua Xu, Yaning Chen, Ling Bai, Yiwen Xu

2016-03-26